# Zr-Rich Eudialyte from the Lovozero Peralkaline Massif, Kola Peninsula, Russia

Taras L. Panikorovskii [1,*], Julia A. Mikhailova [2], Yakov A. Pakhomovsky [2], Ayya V. Bazai [2], Sergey M. Aksenov [1,2], Andrey O. Kalashnikov [2] and Sergey V. Krivovichev [3,4]

[1] Laboratory of Nature-Inspired Technologies and Environmental Safety of the Arctic, Kola Science Centre, Russian Academy of Sciences, 14 Fersman Street, 184200 Apatity, Russia; aks.crys@gmail.com

[2] Kola Science Centre, Geological Institute, Russian Academy of Sciences, 14 Fersman Street, 184209 Apatity, Russia; mikhailova@geoksc.apatity.ru (J.A.M.); pakhom@geoksc.apatity.ru (Y.A.P.); bazai@geoksc.apatity.ru (A.V.B.); kalashnikov@geoksc.apatity.ru (A.O.K.)

[3] Nanomaterials Research Centre of Kola Science Centre, Russian Academy of Sciences, 14 Fersman Street, 184209 Apatity, Russia; s.krivovichev@ksc.ru

[4] Department of Crystallography, Institute of Earth Sciences, Saint–Petersburg State University, University Emb. 7/9, 199034 St. Petersburg, Russia

* Correspondence: t.panikorovskii@ksc.ru

**Abstract:** The Lovozero peralkaline massif (Kola Peninsula, Russia) has several deposits of Zr, Nb, Ta and rare earth elements (REE) associated with eudialyte-group minerals (EGM). Eudialyte from the Alluaiv Mt. often forms zonal grains with central parts enriched in Zr (more than 3 apfu) and marginal zones enriched in REEs. The detailed study of the chemical composition (294 microprobe analyses) of EGMs from the drill cores of the Mt. Alluaiv-Mt. Kedykvyrpakhk deposits reveal more than 70% Zr-enriched samples. Single-crystal X-ray diffraction (XRD) was performed separately for the Zr-rich (4.17 Zr apfu) core and the REE-rich (0.54 REE apfu) marginal zone. It was found that extra Zr incorporates into the octahedral M1A site, where it replaces Ca, leading to the symmetry lowering from $R\bar{3}m$ to $R32$. We demonstrated that the incorporation of extra Zr into EGMs makes the calculation of the eudialyte formula on the basis of Si + Al + Zr + Ti + Hf + Nb + Ta + W = 29 apfu inappropriate.

**Keywords:** eudialyte; eudialyte-group minerals; Lovozero alkaline massif; zirconium; Arctic; mineralogy

## 1. Introduction

Eudialyte-group minerals (EGMs) are trigonal Na-Zr-Ca cyclosilicates, which usually host Mn, Fe, Sr, REE, Y, Cl, F, $CO_3$, $H_2O$, etc. [1]. The chemical composition of eudialyte-group minerals varies in a wide range and strongly depends on the composition of the mineral-forming media. Therefore, EGMs may be considered as geochemical indicators of magmatic crystallization conditions. EGMs may experience secondary transformation due to the ion-exchange processes according to various substitution schemes [2–6]. The possibility of post-crystallization changes results in a wide range of chemical compositions [7] and different space groups [8], which gave rise to the 31 mineral species according to the International Mineralogical Association (IMA) list of approved minerals [9].

EGMs are rock-forming or typical accessory minerals for different types of rocks in the worldwide peralkaline massifs [10]. They have been found in all rock types of the Lovozero peralkaline massif (Kola Peninsula, Russia) [3]. During the evolution of the Lovozero massif, EGMs crystallized at all stages; they indicate the changes in melt composition, oxygen fugacity and temperature. Five new EGM species were discovered in the Lovozero massif (alluaivite, dualite, ikranite, sergevanite, voronkovite), and it seems likely that more new species are about to be discovered [11–14].

EGMs attract global interest as a prospective source of Zr, Nb, Ta and rare-earth elements (REE) [15]. Its significant deposits are located in Pajarito (New Mexico, USA) [16], Lovozero (Kola Peninsula, Russia) [17], Ilimaussaq (South Greenland) [18], Mont Saint-Hilaire (Canada) [19] and Norra Kärr (Sweden) [20]. The relative ease of extraction of EGMs by magnetic separation [21] and the significant share of heavy rare-earth elements (HREE) makes the exploration of EGM deposits economically feasible [22]. Eudialyte is easily decomposable in acids, but the formation of amorphous silica gel reduces the level of the REE and zirconium extraction into solution [23]. The economic viability has yet to be demonstrated under industry scales, though several promising multi-step leaching techniques were developed [24,25].

The knowledge of the structural state of the critical elements (REEs and Zr) in EGMs is important for the industry extraction technologies [15]. The detailed studies of incorporation of REE into eudialyte structure were reported in [15]. In contrast, much less is known about the Zr-rich EGMs. According to the literature data, the Zr-content in EGM from most deposits does not exceed 3 apfu. The $ZrO_2$ and $REE_2O_3$ content vary in the range (wt.%): 11.40–12.07; 2.00–3.30 for North Qôroq centre, South Greenland; 10.35–11.48; 0.39–10.15 for Mont Saint-Hilaire, Quebec, Canada; 9.85–10.82; 0.15–0.52 for Gardiner complex, East Greenland; 10.90–11.20, 1.15–12.12 for Ilímaussaq, Southern Greenland; 0.15–5.00 ($ZrO_2$) for Tanbreez, South Greenland [7,26–28]. Meanwhile, most of the EGM samples (70%) from the rocks of the Lovozero massif are hyperzirconium, i.e., their Zr content exceeds 3 apfu [3]. According to our data, the $ZrO_2$ content in EGM from Lovozero massif ranges from 6.32 to 17.14 wt.% and $REE_2O_3$ varies in the range 0.35–5.92 wt.%.

In this work, we present the results of studying the chemical composition and crystal structure of Zr-rich (up to 4.17 apfu Zr) and REE-enriched zones (up to 0.30 apfu REEs) of EGM samples. The EGM samples were taken from the drill cores of the Mt. Alluaiv-Kedykvyrpakhk eudialyte deposit, the north-west part of Lovozero massif, Kola Peninsula, Russia (Figure 1). We discussed the main substitution mechanisms associated with the extra Zr incorporation into EGMs based on the data of 294 microprobe analyses in the view of general petrology and geochemistry of the Lovozero massif [24].

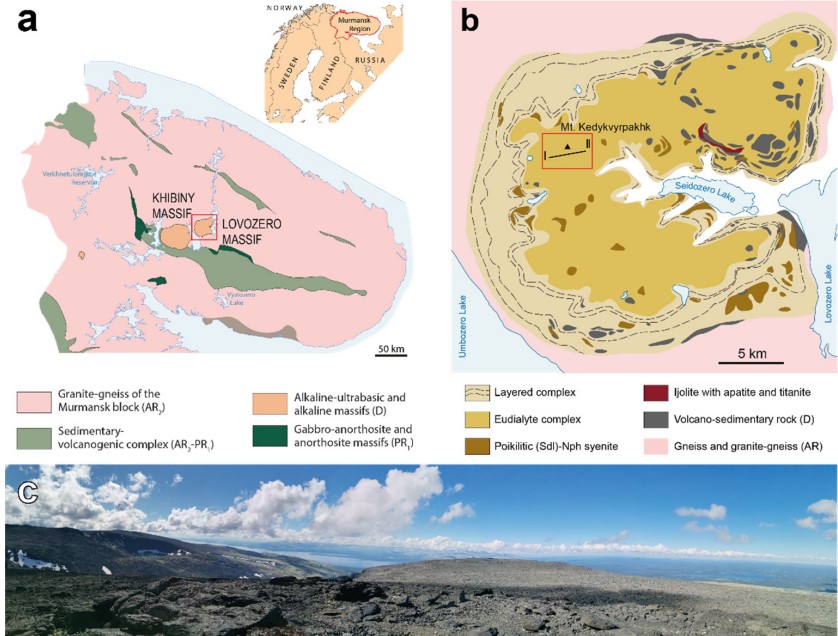

**Figure 1.** (**a**) General geological scheme of Kola Peninsula and location of the Lovozero massif within the red square; (**b**) geological scheme of the Lovozero massif and the location of Alluaiv-Kedykvyrpakhk eudialyte deposit within the red square, based on [3]; (**c**) The view from the top of the Kedykverpakhk Mt. to the Umbozero Lake side. The black line I–II represents the section of the Alluaiv mine.

## 2. Geological Setting

The Lovozero peralkaline massif is a layered laccolith-like intrusion with an area of 650 km$^2$. It intruded into the Archean granite-gneisses [29] (Figure 1a) and was emplaced at 370 ± 7 Ma by Rb–Sr dating [30]. The Lovozero massif (Figure 1b) is composed of three major units: Layered, Eudyalite and Poikilitic complexes [29,31–33].

Layered complex (77% of massif volume) is composed of alternated sub-horizontal layers or "rhythms". Each rhythm is a sequence of rocks (from top to bottom): lujavrite-foyaite-urtite or lujavrite-foyaite. The transitions between the listed rocks within the rhythm are gradual, and the boundaries between the rhythms are sharp and are often marked by pegmatites. Lujavrite is a coarse- to medium-grained mesocratic nepheline syenite with a trachytoid texture (laths of alkali feldspar are oriented parallel to each other). Foyaite is a coarse- to medium-grained massive leucocratic nepheline syenite, and urtite is an almost monomineralic nepheline rock.

The Eudialyte complex (18% of massif volume) overlies the Layered complex and consists of lujavrite rock enriched in EGM (eudialyte lujavrite). Lenses and sheet-like bodies of foyaite, as well as porphyritic and fine-grained nepheline syenites, are irregularly located among eudialyte lujavrite.

Poikilitic complex (5% of the massif volume) consists of poikilitic feldspathoid (nepheline, sodalite and vishnevite) syenites. The rocks of this complex form lens, sheet-like bodies or irregularly shaped bodies that are located inside the rocks of Layered and Eudialyte complexes.

There are a large number of xenoliths of Devonian volcaniclastic rocks [32,34], both unaltered and intensely metasomatized (fenitized) found among the rocks of the Layered and Eudialyte complexes.

The Alluaiv-Kedykvyrpakhk eudialyte deposit is part of the Eudialyte complex. The ore consists of a eudialyte lujavrite, whose EGM content can reach 80% of the rock volume. The interest in the Lovozero EGMs as a source of Zr and REEs is due to the gigantic amounts of these minerals, their comparatively easy ore processing and the possibility of product recovery [35].

## 3. Materials and Methods

Two EGM samples (LV-153/178 and LV-117/226) from the most common rocks of the Eudialyte complex with high Zr and REE content were selected for chemical and single-crystal XRD studies.

The LV-153/178 sample was selected from medium-grained eudialyte lujavrite. The rock-forming minerals are microcline-perthite (30 modal%), nepheline 15%, aegirine 20%, magnesioarfvedsonite 10% and eudialyte 25%. Accessory minerals are lamprophyllite, sphalerite and natrolite. Microcline-perthite laths are oriented sub-parallel, and the space between them is filled with subhedral to anhedral nepheline crystals and clusters of long-prismatic crystals of aegirine and anhedral grains of magnesioarfvedsonite. Eudialyte grains are spatially confined to clusters of mafic minerals (Figure 2a). Chemically zoned EGM grains (up to 1 cm in length, Figure 2b) with REE-enriched rims and Zr-rich cores are typical for eudialyte lujavrite. The composition and crystal structure of the REE-rich rim from the LV-153/178 sample were studied (Figure 2a).

The LV-117/226 sample was selected from porphyritic nepheline syenite. Fine-grained mass consists of euhedral albite (25 modal%), anhedral microcline 20%, anhedral nepheline 15%, euhedral aegirine 20% and anhedral or poikilitic magnesioarfvedsonite 5%. Phenocrysts of microcline-perthite and nepheline contain numerous inclusions of aegirine, albite and sometimes EGM (Figure 2c). EGMs form small individual grains (up to 200 μm) in the coarse-grained matrix. In the porphyritic nepheline syenite, EGMs form chemically homogeneous rounded grains with exceptionally high ZrO$_2$ content (up to 17.14 ZrO$_2$ wt.%) in the crystal cores. The most Zr-rich central part of the LV-117/226 sample indicated in Figure 2d was investigated by SC XRD analysis.

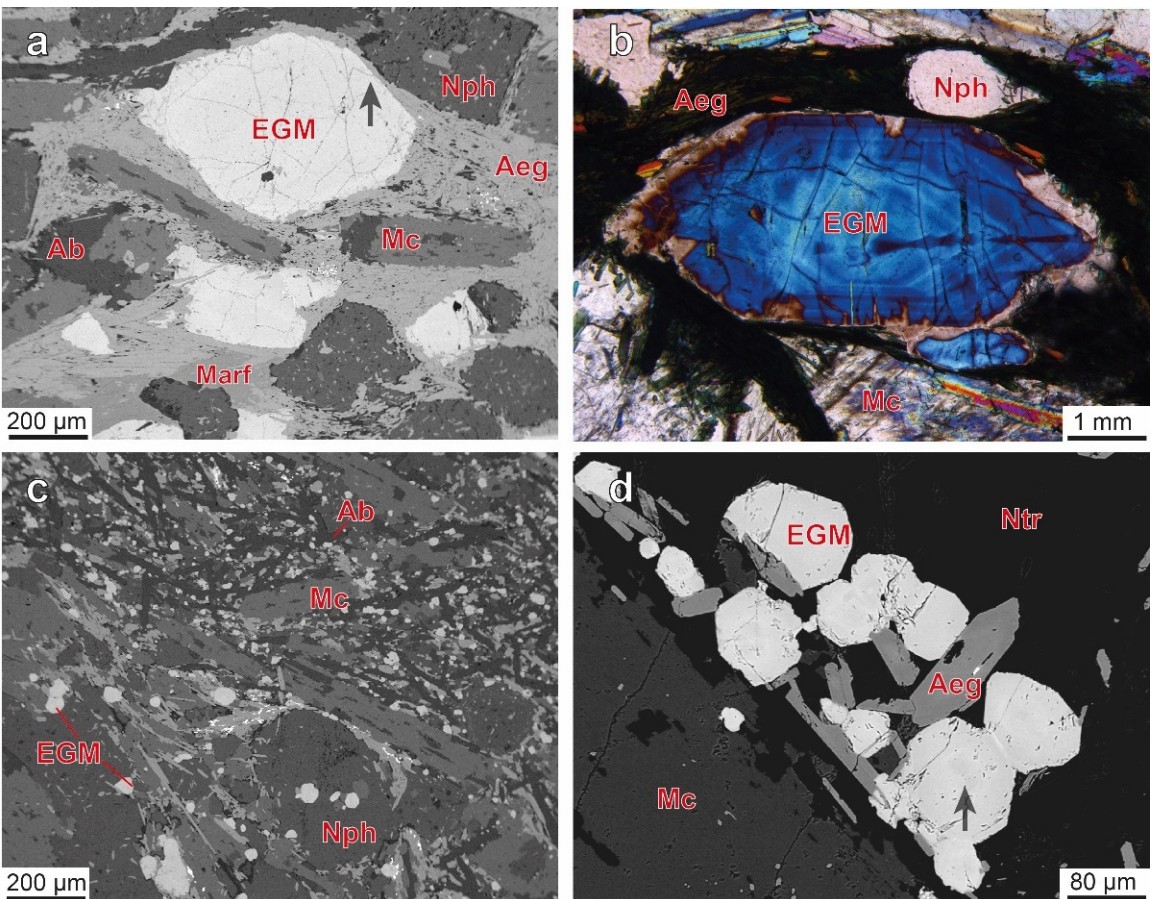

**Figure 2.** (**a**) Backscattered electron (BSE) image of eudialyte lujavrite rock (Sample LV-153/178). Black arrow indicates point were selected crystal part for SC XRD analysis; (**b**) optically zonal EGM grain from LV-153/178 sample under crossed polars; (**c**) BSE image of porphyritic nepheline syenite (LV-117/226 sample); (**d**) Zr-enriched EGM from porphyritic nepheline syenite (LV-117/226 sample). Associated minerals are albite (Ab), aegirine (Aeg), magnesio-arfvedsonite (Marf), microcline (Mc) nepheline (Nph) and natrolite (Ntr).

Polished sections were analyzed using Leica M205 polarizing stereomicroscope equipped by Leica DFC295 camera.

The thin polished sections were analyzed using the scanning electron microscope LEO-1450 (Carl Zeiss Microscopy, Oberkochen, Germany), with energy-dispersive microanalyzer Quantax 200 (Bruker, Billerica, MA, USA) to obtain BSE (Back Scattered Electron, Carl Zeiss Microscopy, Oberkochen, Germany) images and pre-analyze all detected minerals.

The composition of EGMs was determined with a Cameca MS-46 microprobe (Cameca, Gennevilliers, France) operated in a WDS mode at 22 kV and 20–40 nA and a beam diameter of 10 μm with counting times of 20 s (for a peak) and $2 \times 10$ s (for background before and after the peak), with 5–10 counts for every element in each point. The following standards were used: lorenzenite (Na, Ti), pyrope (Al), wollastonite (Si, Ca), fluorapatite (P), $F_{10}S_{11}$ (Fe, S), atacamite (Cl), wadeite (K), metallic V, $MnCO_3$ (Mn), hematite (Fe), celestite (Sr), $ZrSiO_4$ (Zr), metallic Nb, barite, $LaCeS_2$ (La, Ce), $LiPr(WO_4)_2$ (Pr), $LiNd(MoO_4)_2$ (Nd), $LiSm(MoO_4)_2$ (Sm), metallic Hf and Ta, thorite (Th) and metallic U.

Chemical contents (in atoms per formula unit, apfu) were calculated with the MINAL program of D. Dolivo-Dobrovolsky [36]. Statistical analyses were carried out with the STATISTICA 8.0 software (Statsoft company, Dell, Round Rock, TX, USA) [37].

The crystal structures of the EGMs were studied using a Rigaku/Oxford Diffraction XtaLAB Supernova diffractometer at room temperature. More than a hemisphere of diffraction data were collected using CuKα-radiation and scanning along ω with a step of 1° and the exposure time of 10 s. Empirical absorption correction was applied in the

CrysAlisPro [38] program complex using spherical harmonics, implemented in the SCALE3 ABSPACK scaling algorithm. The unit cells were refined by the least-squares methods. The structure was refined in the SHELX program [39]. The crystal structure was drawn using the Diamond program [40]. Occupancies of the cation sites were calculated from the experimental site-scattering factors in accordance with the empirical chemical composition.

The crystal structure of the eudialyte was first reported in 1971 by Golyshev et al. [41] and Giuseppetti et al. [42], who proposed three possible space groups, $R\overline{3}m$, $R3m$ and $R32$, but refined the structure in the $R\overline{3}m$ space group only. According to the systematic investigation, EGMs may crystallize in the $R\overline{3}m$, $R3m$ or $R3$ space groups [7]. The crystal structures of the LV-117/226 and LV-153/178 samples were refined in the $R\overline{3}m$, $R3m$, $R32$ and $R\overline{3}$ space groups.

For the LV-153/178 sample, the lowering of symmetry did not improve the $R$-value and the $R\overline{3}m$ space group was chosen for the final refinement. The crystal structure of LV-153/178 was refined to $R_1 = 0.037$ for 1300 independent reflections with $F^2 > 4\sigma(F^2)$, respectively, with the site nomenclature following the IMA recommendations [1].

The crystal structure of the LV-117/226 sample was initially refined in the $R\overline{3}m$ space group to $R_1 = 0.080$. The refinement in the $R\overline{3}$ space group resulted in $R_1 = 0.083$ and physically unrealistic displacement parameters for the M1A,B sites. The refinement in the $R3m$ space group to $R_1 = 0.072$ did not sufficiently improve the structure model. The best refinement was performed in the $R32$ space group with $R_1 = 0.034$ for 2591 independent reflections with $F^2 > 4\sigma(F^2)$.

The crystal structure data are deposited in the CCDC under the entry numbers 2082760-2082761.

Geostatistical studies, interpolation, and 3D modeling were conducted by the MINE-FRAME 8 program (Mining Institute of Kola Science Centre, Russian Academy of Sciences, https://www.mineframe.ru, accessed on 15 July 2021) and Micromine 2016.1 (Micromine Pty Ltd., Pert, Australia, https://www.micromine.com; commercial license, accessed on 15 July 2021). Interpolation was carried out by an inverse distance method.

## 4. Results

### 4.1. Chemical Composition

The calculation of the EGMs formulas was performed according to recommendations by Johnsen and Grice [7]. The general EGM formula can be written as:

$$N(1)_3N(2)_3N(3)_3N(4)_3N(5)_3M(2)_{3\text{-}6}M(3)M(4)[M(1)_6Z_3(Si_{24}O_{72})]O'_{4\text{-}6\times2},$$

where $N(1\text{–}5)$ = Na, $H_3O^+$, K, Sr, Ln, Y, Ba, Mn and Ca; $M(1)$ = Ca, Mn, Ln, Na, Sr and Fe; $M(2)$ = Fe, Mn, Na, Zr, Ta, Ti, K, Ba and $H_3O^+$; $M(3)$ and $M(4)$ = Si, Nb, Ti, W and Na; $Z$ = Zr, Ti, Nb; $O'$ = O, (OH) and $H_2O$; $X$ = Cl, F, $H_2O$, OH, $CO_3$ and $SO_4$.

Table 1 provides microprobe results for structurally investigated EGMs. The calculation based on Si + Al + Zr + Ti + Hf + Nb + Ta + W = 29 apfu shows the excess of Zr (more than 3 apfu) in all investigated samples.

**Table 1.** Microprobe analyses of the EGMs studied by single-crystal XRD analysis.

| Sample | 117/226 * Core | 153/178 Rim | 156/36 Core | 156/36 Rim | 169/143 Core | 169/143 Rim | 117/226 Core | 153/178 Rim |
|---|---|---|---|---|---|---|---|---|
| $SiO_2$ | 48.85 | 51.01 | 52.60 | 52.12 | 49.51 | 51.59 | 48.85 | 51.01 |
| $TiO_2$ | 0.54 | 0.58 | 0.59 | 0.46 | 0.55 | 0.52 | 0.54 | 0.58 |
| $Al_2O_3$ | 0.24 | 0.16 | 0.35 | 0.23 | 0.16 | 0.07 | 0.24 | 0.16 |
| FeO | 1.51 | 2.23 | 2.96 | 2.87 | 1.97 | 2.00 | 1.51 | 2.23 |
| MnO | 3.11 | 2.19 | 2.42 | 2.68 | 3.09 | 3.08 | 3.11 | 2.19 |
| MgO | b.d | 0.07 | 0.10 | 0.08 | b.d | b.d | b.d | 0.07 |
| CaO | 6.21 | 7.99 | 7.33 | 8.17 | 7.90 | 8.32 | 6.21 | 7.99 |
| $Na_2O$ | 8.43 | 14.61 | 5.43 | 5.15 | 17.23 | 16.58 | 8.43 | 14.61 |
| $K_2O$ | 0.24 | 0.30 | 0.59 | 0.76 | 0.23 | 0.17 | 0.24 | 0.30 |
| SrO | 2.74 | 1.83 | 2.04 | 2.59 | 2.37 | 1.53 | 2.74 | 1.83 |
| $Y_2O_3$ | 0.41 | b.d | 0.61 | 0.66 | b.d | b.d | 0.41 | b.d |
| $ZrO_2$ | 17.14 | 12.78 | 14.04 | 13.09 | 11.88 | 11.08 | 17.14 | 12.78 |
| $Nb_2O_5$ | 0.36 | 0.87 | 0.60 | 0.86 | 1.03 | 0.55 | 0.36 | 0.87 |
| BaO | 0.19 | b.d | 0.13 | 0.23 | 0.34 | 0.47 | 0.19 | b.d |
| $La_2O_3$ | 0.30 | 0.24 | 0.26 | 0.35 | 0.28 | 0.41 | 0.30 | 0.24 |
| $Ce_2O_3$ | 0.54 | 0.83 | 0.59 | 0.84 | 0.84 | 1.20 | 0.54 | 0.83 |
| $Nd_2O_3$ | 0.39 | 0.35 | 0.30 | 0.32 | 0.35 | 0.47 | 0.39 | 0.35 |
| $HfO_2$ | 0.22 | 0.00 | 0.65 | 0.31 | b.d | b.d | 0.22 | 0.00 |
| Cl | 1.56 | 1.44 | 1.41 | 1.53 | 1.56 | 0.74 | 1.56 | 1.44 |
| $H_2O$ † | 3.69 | 2.03 | | | | | 3.69 | 2.03 |
| Sum | 97.69 | 99.51 | 93.00 | 93.29 | 99.27 | 98.76 | 97.69 | 99.51 |
| Formula based on $\Sigma$(Si + Al + Zr + Ti + Hf + Nb + Ta + W) normalized to 29 apfu | | | | | | | ‡ | § |
| $Si^{4+}$ | 24.37 | 24.39 | 25.11 | 25.36 | 25.47 | 25.92 | 24.85 | 24.54 |
| $Ti^{4+}$ | 0.20 | 0.22 | 0.21 | 0.17 | 0.21 | 0.20 | 0.21 | 0.21 |
| $Al^{3+}$ | 0.14 | 0.09 | 0.20 | 0.13 | 0.10 | 0.04 | 0.15 | 0.09 |
| $Fe^{2+}$ | 0.63 | 0.93 | 1.18 | 1.17 | 0.85 | 0.84 | 0.64 | 0.90 |
| $Mn^{2+}$ | 1.31 | 0.92 | 0.98 | 1.10 | 1.34 | 1.31 | 1.34 | 0.89 |
| $Mg^{2+}$ | - | 0.05 | 0.07 | 0.06 | - | - | - | 0.05 |
| $Ca^{2+}$ | 3.32 | 4.26 | 3.75 | 4.26 | 4.35 | 4.48 | 3.39 | 4.12 |
| $Na^+$ | 8.15 | 14.10 | 5.03 | 4.86 | 17.19 | 16.16 | 8.31 | 13.63 |
| $K^+$ | 0.15 | 0.19 | 0.36 | 0.47 | 0.15 | 0.11 | 0.16 | 0.18 |
| $Sr^{2+}$ | 0.79 | 0.53 | 0.56 | 0.73 | 0.71 | 0.44 | 0.81 | 0.51 |
| $Y^{3+}$ | 0.11 | - | 0.16 | 0.17 | - | - | 0.11 | - |
| $Zr^{4+}$ | 4.17 | 3.10 | 3.27 | 3.11 | 2.98 | 2.71 | 4.25 | 3.00 |
| $Nb^{5+}$ | 0.08 | 0.20 | 0.13 | 0.19 | 0.24 | 0.12 | 0.08 | 0.19 |
| $La^{3+}$ | 0.05 | 0.04 | 0.05 | 0.06 | 0.05 | 0.08 | 0.05 | 0.04 |
| $Ce^{3+}$ | 0.10 | 0.15 | 0.10 | 0.15 | 0.16 | 0.22 | 0.10 | 0.15 |
| $Nd^{3+}$ | 0.07 | 0.06 | 0.05 | 0.06 | 0.06 | 0.08 | 0.07 | 0.06 |
| $Hf^{4+}$ | 0.03 | - | 0.09 | 0.04 | - | - | 0.03 | |
| $H^+$ | 12.28 | 6.74 | - | - | - | - | 12.52 | 6.51 |
| $Cl^-$ | 1.32 | 1.22 | 1.14 | 1.26 | 1.36 | 0.63 | 1.34 | 1.17 |

* Sample LV-117/226 also contain 0.25 wt.% $SO_3$ or 0.1 S apfu, 0.19 wt.% BaO or 0.04 apfu, 0.33 wt.% $Pr_2O_3$ or 0.06 Pr apfu, 0.33 wt.% $Sm_2O_3$ or 0.06 Sm apfu, 0.11 wt.% $Gd_2O_3$ or 0.02 Gd apfu. † $H_2O$ content calculated from the XRD data. ‡ Formulae normalized on Si content from structural data (Si = 24.85 for 117/226). § (Si = 24.54 for 153/178). b.d—below detection limit.

### 4.2. Crystal Structure: General Scheme

The crystal structure of EGMs can be described as a stacking of complex *ZTMT* modules perpendicular to [001] with $t_q = 1/2a + 1/3c$ (Figure 3a) [43]. Each module contains four *Z*, *T*, *M* and *T* layers based on *T*-, *Z*- and *M*-polyhedra. The tetrahedral *T*-layers (Figure 3d) have identical topologies and consist of the $[Si_3O_9]^{6-}$ and $[Si_9O_{27}]^{18-}$ rings linked by sharing corners with the *N*4 distorted octahedra (Figure 3c) [41,42]. The nine-membered rings may be centered by additional *M*4B tetrahedra occupied by Si or be vacant. Ordering of the additional Si-centered tetrahedra may result in the doubling of the *c* parameter~60 Å [11]. The *Z*-layers are sandwiched between two *T*-layers and composed of $ZrO_6$ octahedra and N1A,B polyhedra. The *M*-layer (Figure 3b) consists of octahedral

6- and 9-membered rings based upon $M1O_6$ octahedra or alternating $M1O_6$ and $M2O_6$ octahedra (that can be replaced by square pyramids and squares). One-half of the nine-membered rings are usually centered by additional $M4A$ octahedra. Two different types of the *M*-layers were observed in several "megaeudialytes" [44,45]. The $Cl^-$ and $OH^-$ anions are located in the cavities of the *M* layer and may reach up to 2 apfu. The eudialyte-type *MT*-framework usually contains significant numbers of split and low-occupied sites (*N*1A,B, *M*2A,B, *M*3 and *M*4).

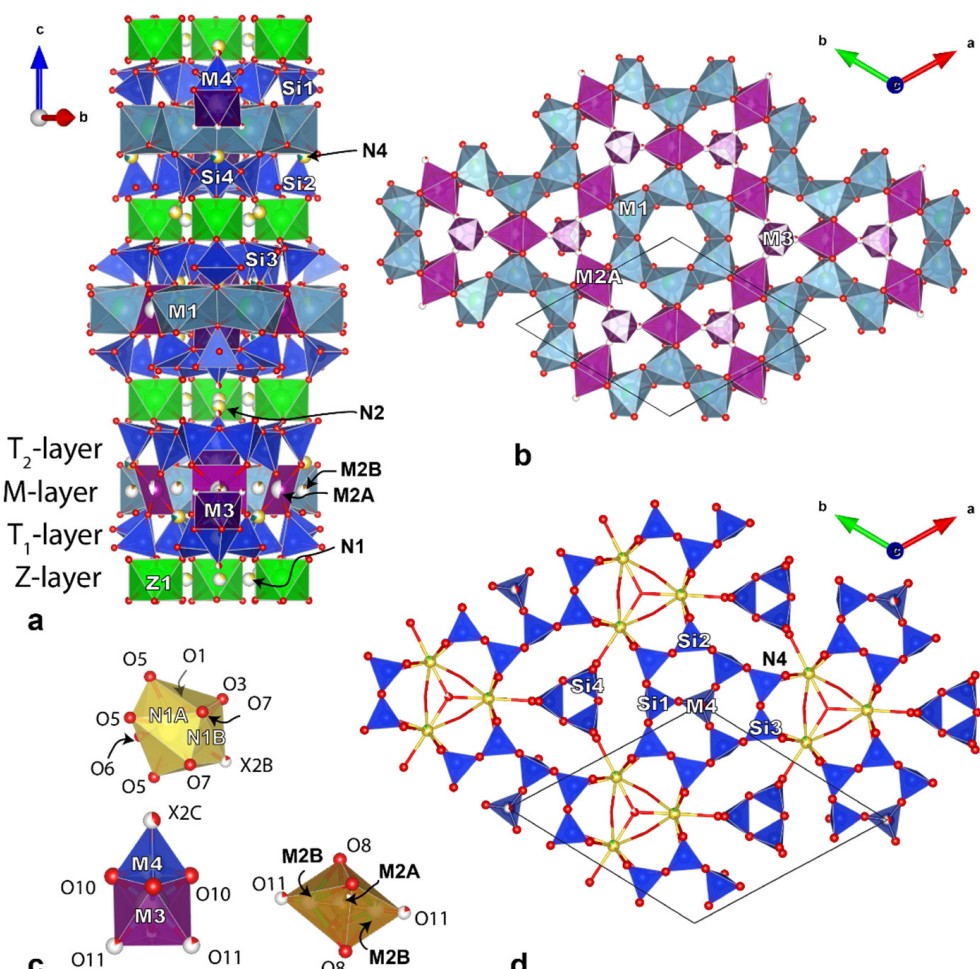

**Figure 3.** The $R\bar{3}m$ model of the 12-layered ($c \sim 30$ Å) EGM crystal structure: (**a**) general view, (**b**) *M*-layer projected on the plane (001) (**c**) split N1A,B, *M*4A,B and *M*2A,B sites (**d**) *T*-layer projected on the plane (001).

Crystal data, data collection information and structure refinement details are provided in Table 2; atom coordinates, site occupancies and isotropic displacement parameters are provided in Tables 3 and 4 for LV-153/178 and LV-117/226, respectively. Selected bond lengths and anisotropic displacement parameters are given in Tables S1–S4.

**Table 2.** Crystal data and structure refinement for the EGMs from samples LV-153/178 and LV-117/226.

| Sample | LV-153/178 | LV-117/226 |
|---|---|---|
| Temperature/K | 296.15 | 293(2) |
| Crystal system | trigonal | trigonal |
| Space group | $R\bar{3}m$ | $R32$ |
| $a$, Å | 14.2248(7) | 14.2081(4) |
| $c$, Å | 30.3453(15) | 30.3723(7) |
| Volume, Å$^3$ | 5317.6(6) | 5309.8(3) |
| Z | 3 | 3 |
| $\rho_{calc}$, g/cm$^3$ | 2.876 | 2.897 |
| $\mu$, mm$^{-1}$ | 2.113 | 2.711 |
| F(000) | 4472.0 | 4495.0 |
| Crystal size, mm$^3$ | $0.12 \times 0.11 \times 0.08$ | $0.14 \times 0.11 \times 0.05$ |
| Radiation | Cu$K\alpha$ ($\lambda$ = 1.54184) | Mo$K\alpha$ ($\lambda$ = 0.71073) |
| 2$\Theta$ range for data collection, ° | 7.746 to 143.09 | 5.734 to 54.978 |
| Index ranges | $-17 \leq h \leq 13, -10 \leq k \leq 17, -36 \leq l \leq 36$ | $-13 \leq h \leq 12, -5 \leq k \leq 18, -39 \leq l \leq 29$ |
| Reflections collected | 4917 | 4362 |
| Independent reflections | 1300 ($R_{int}$ = 0.0292, $R_{sigma}$ = 0.0259) | 2591 ($R_{int}$ = 0.0186, $R_{sigma}$ = 0.0313) |
| Data/restraints/parameters | 1300/0/152 | 2591/0/247 |
| Goodness-of-fit on F$^2$ | 1.088 | 1.066 |
| Final $R$ indexes [I $\geq$ 2$\sigma$ (I)] | $R_1$ = 0.0375, w$R_2$ = 0.0903 | $R_1$ = 0.0344, w$R_2$ = 0.0829 |
| Final $R$ indexes [all data] | $R_1$ = 0.0418, w$R_2$ = 0.0918 | $R_1$ = 0.0374, w$R_2$ = 0.0847 |
| Largest diff. peak/hole/e Å$^{-3}$ | 0.85/$-0.80$ | 1.40/$-0.71$ |
| Flack parameter | n.d. | 0.50(3) |

**Table 3.** Atomic coordinates, occupancies and equivalent isotropic displacement parameters LV-153/178 EGM sample.

| Site | Occupancy | $x/a$ | $y/b$ | $z/c$ | $U_{ani}$ |
|---|---|---|---|---|---|
| M1 | $Ca_{0.91}Ce_{0.09}$ | $2/3$ | 0.59499(7) | $5/6$ | 0.0156(4) |
| M2A | $Fe_{0.66}\square_{0.34}$ | $1/2$ | $1/2$ | $1/2$ | 0.080(2) |
| M2B | $\square_{0.97}Zr_{0.03}$ | 0.432(2) | 0.568(2) | 0.5022(7) | 0.080(2) |
| M3 | $\square_{0.67}Nb_{0.33}$ | $2/3$ | $1/3$ | 0.45507(15) | 0.0419(15) |
| M4 | $\square_{0.73}Si_{0.27}$ | $2/3$ | $1/3$ | 0.4118(3) | 0.011(4) |
| Z1 | Zr | $1/3$ | $1/6$ | $2/3$ | 0.0124(2) |
| N1A | $Na_{0.75}\square_{0.25}$ | 0.5568(2) | 0.4432(2) | 0.67991(13) | 0.0259(9) |
| N1B | $\square_{0.75}Na_{0.25}$ | 0.5863(7) | 0.4137(7) | 0.6637(4) | 0.032(3) |
| N4 | $Na_{0.91}Sr_{0.09}$ | 0.4592(2) | 0.22959(12) | 0.54912(9) | 0.0473(12) |
| N5 | $Na_{0.83}\square_{0.17}$ | $2/3$ | $1/3$ | 0.5957(3) | 0.061(3) |
| Si1 | Si | 0.73712(7) | 0.47424(13) | 0.75170(5) | 0.0135(4) |
| Si3 | Si | 0.79207(7) | 0.58413(14) | 0.42439(5) | 0.0179(4) |
| Si5 | Si | 0.39705(9) | 0.38977(9) | 0.59703(3) | 0.0132(3) |
| O1 | O | 0.60481(17) | 0.39519(17) | 0.75584(15) | 0.0215(10) |
| O2 | O | 0.78059(19) | 0.5612(4) | 0.79000(13) | 0.0258(11) |
| O3 | O | 0.7617(2) | 0.5234(4) | 0.70273(14) | 0.0277(11) |
| O4 | O | 0.48607(19) | 0.51393(19) | 0.80471(13) | 0.0196(9) |
| O5 | O | 0.3882(3) | 0.4339(3) | 0.72670(10) | 0.0301(8) |
| O6 | O | 0.4890(2) | 0.5110(2) | 0.61392(14) | 0.0238(10) |
| O7 | O | 0.4074(3) | 0.3036(3) | 0.62716(11) | 0.0293(8) |
| O8 | O | 0.4108(3) | 0.3742(3) | 0.54553(9) | 0.0215(7) |
| O9 | $O_{0.64}\square_{0.36}$ | $2/3$ | $1/3$ | 0.5454(6) | 0.051(4) |
| O10 | O | 0.7271(2) | 0.4543(5) | 0.4257(3) | 0.0605(19) |
| O11 | $\square_{0.67}(H_2O)_{0.33}$ | 0.3920(11) | 0.6080(11) | 0.5055(8) | 0.017(5) |
| X1 | $\square_{0.77}Cl_{0.33}$ | 0.5932(5) | 0.4068(5) | 0.3447(4) | 0.122(6) |
| X2A | $\square_{0.94}Cl_{0.06}$ | $2/3$ | $1/3$ | 0.501(2) | 0.051(4) |
| X2B | $\square_{0.84}(H_2O)_{0.16}$ | $2/3$ | $1/3$ | 0.605(6) | 0.050 |
| X2C | $\square_{0.73}(H_2O)_{0.37}$ | $2/3$ | $1/3$ | 0.3593(10) | 0.045(7) |

**Table 4.** Atomic coordinates, occupancies and equivalent isotropic displacement parameters LV-117/226 EGM sample.

| Site | Occupancy | $x/a$ | $y/b$ | $z/c$ | $U_{ani}$ |
|---|---|---|---|---|---|
| M1A | $Ca_{0.61}Zr_{0.39}$ | 0.58616(14) | $2/3$ | $1/6$ | 0.0160(5) |
| M1B | $Ca_{0.88}Ce_{0.12}$ | 0.06533(14) | $2/3$ | $1/6$ | 0.0158(6) |
| M2A | $Mn_{0.53}\square_{0.47}$ | 0.8287(4) | $2/3$ | $1/6$ | 0.085(3) |
| M2B | $\square_{0.89}Fe_{0.11}$ | 0.7684(12) | 0.5478(8) | 0.1694(3) | 0.026(3) |
| M3 | $\square_{0.67}Ti_{0.33}$ | $2/3$ | $1/3$ | 0.2143(2) | 0.0283(15) |
| M4 | $\square_{0.58}Si_{0.42}$ | $2/3$ | $1/3$ | 0.25530(17) | 0.0097(19) |
| Z1 | Zr | $2/3$ | 0.83849(8) | $1/3$ | 0.01012(19) |
| N1 | $Na_{0.91}Sr_{0.09}$ | 0.1037(3) | 0.8933(3) | 0.21659(10) | 0.0626(14) |
| N3 | $Na_{0.75}\square_{0.25}$ | 0.8861(6) | 0.7787(4) | 0.34644(12) | 0.0271(9) |
| N4 | $\square_{0.75}Na_{0.25}$ | 0.9126(18) | 0.8343(12) | 0.3328(4) | 0.031(3) |
| Si1 | Si | 0.91643(12) | 0.4573(2) | 0.24247(5) | 0.0134(3) |
| Si2 | Si | 0.93912(18) | 0.67238(17) | 0.26343(8) | 0.0114(4) |
| Si3 | Si | 0.72728(16) | 0.67626(17) | 0.26376(8) | 0.0103(4) |
| Si4 | Si | 0.47422(12) | 0.7423(2) | 0.24804(4) | 0.0131(3) |
| O1 | O | 0.9596(5) | 0.3915(4) | 0.2738(2) | 0.0207(14) |
| O2 | O | 0.7865(4) | 0.3809(8) | 0.2397(3) | 0.060(2) |
| O3 | O | 0.9713(3) | 0.4878(6) | 0.19551(12) | 0.0184(8) |
| O4 | O | 0.9436(7) | 0.5613(6) | 0.2732(2) | 0.0319(17) |
| O5 | O | 0.8184(5) | 0.6455(3) | 0.28026(13) | 0.0195(8) |
| O6 | O | 0.0253(5) | 0.7666(6) | 0.2940(2) | 0.0282(15) |
| O7 | O | 0.7393(5) | 0.7057(5) | 0.2127(2) | 0.0186(13) |
| O8 | O | 0.7403(6) | 0.7732(5) | 0.2943(2) | 0.0227(14) |
| O9 | O | 0.5600(4) | 0.7863(7) | 0.20933(13) | 0.0292(12) |
| O10 | O | 0.3993(6) | 0.6091(6) | 0.24431(14) | 0.0214(9) |
| O11 | O | 0.5248(4) | 0.7687(8) | 0.29659(14) | 0.0302(12) |
| O12 | O | 0.9561(5) | 0.7014(5) | 0.2124(2) | 0.0214(14) |
| O13 | $\square_{0.67}(H_2O)_{0.33}$ | 0.729(4) | 0.457(3) | 0.1714(9) | 0.030(10) |
| O14 | $\square_{0.58}(H_2O)_{0.42}$ | $2/3$ | $1/3$ | 0.3078(4) | 0.013(5) |
| X1A | $\square_{0.67}(H_2O)_{0.33}$ | 0 | 0 | 0.1633(9) | 0.108(17) |
| X1B | $\square_{0.72}(H_2O)_{0.28}$ | 0 | 0 | 0.3096(14) | 0.10(3) |
| X2A | $\square_{0.22}(H_2O)_{0.78}$ | 0 | 0 | 0.2650(5) | 0.048(5) |
| X2B | $\square_{0.23}(H_2O)_{0.77}$ | 0 | 0 | 0.2089(2) | 0.008(2) |
| X2C | $\square_{0.69}Cl_{0.31}$ | 0.5099(9) | 0.2708(19) | 0.3238(4) | 0.108(7) |

### 4.3. Crystal Structure: LV-153/178

*T-layers*. The Si1, Si3 and Si5 sites are fully populated by Si atoms with the <Si–O> distances in the range 1.594–1.645 Å. The additional tetrahedral M4 site is partially populated owing to the short M3-M4 distance of ~1.30 Å. The refined occupancy of the M4 site is $Si_{0.27}$ or 0.54 Si apfu. The total refined Si content is 24.54 apfu that is less than 25.56 Si apfu calculated from the chemical data. The additional octahedral M3 site has an occupancy of $Nb_{0.33}$ or 0.65 Nb apfu and an average Nb–O distance of 1.808 Å. The nine-coordinated N4 site is populated by Na with a small admixture of Sr (Table 3).

*Z-layers*. The octahedral Z1 sites are fully populated by Zr. The average <Z1–O> distance is 2.071 Å. The N1 site is split into 8-coordinated N1A and 7-coordinated N1B subsites with the refined occupancies of $Na_{0.75}$ and $Na_{0.25}$, respectively, and the <Na–O> distances of 2.644 and 2.659 Å, respectively (Figure 4).

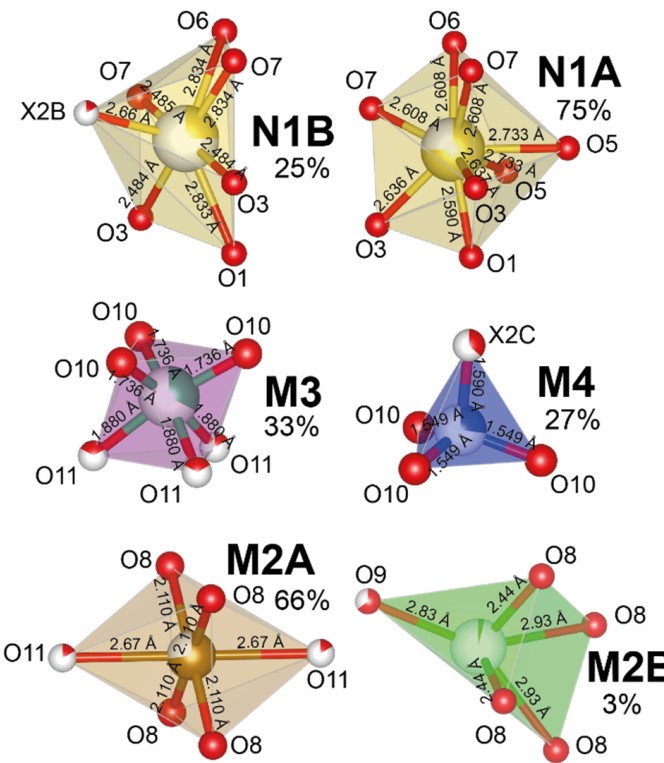

**Figure 4.** The local coordination and occupancy of the split sites in the crystal structure LV-153/178 EGM sample.

*M-layer.* The octahedral M1 site is populated by Ca with a minor admixture of REE that is in agreement with the previously reported XANES data [15]. The refined REE content is 0.54 apfu, which exceeds 0.25 REE apfu derived from the chemical data. The additional electron density may be explained by a possible admixture of Zr. The M1–O distances are in the range 2.316–2.392 Å. The M2 site split into the M2A and M2B sites with the M2A-M2B distance of 1.685 Å. The M2A octahedral site has four short M2A–O8 bonds of 2.110 Å and two long distances of 2.670 Å to the low-occupied O11 site. The refined scattering value for the M2A site is 17.60 e$^-$ corresponds to the refined occupancy of $Fe_{0.66}$. According to [46], the extra Zr-content is associated with the low occupied five-coordinated M2B site (Figure 4) with the refined occupancy of $Zr_{0.03}$.

*X-sites.* In the studied sample, there are low-occupied X1, X2A sites populated by Cl and the X2B and X2C sites populated by $H_2O$ molecules.

The refined formulae for the sample LV-153/178 can be written as:

$$^{N1-5}(Na_{12.87}Sr_{0.55})_{\Sigma13.42}{}^{M1}(Ca_{5.47}REE_{0.53})_{\Sigma6.00}{}^{M2}(Fe^{3+}{}_{1.46}\square_{0.83}Fe^{2+}{}_{0.51}Zr_{0.20})_{\Sigma3.00}{}^{M3}(Nb_{0.65}\square_{0.35})$$
$$^{M4}(Si_{0.54}\square_{0.46})_{\Sigma1.00}{}^{Z1}(Zr_{3.00})Si_{24}O_{72}{}^{X1,2A}Cl_{2.11}{}^{X2A,B}(H_2O)_{3.36}$$

The empirical formulae based on $Si_{24.54}$ may be written as:

$$^{N1-5}(Na_{13.63}Sr_{0.51}K_{0.18})_{\Sigma14.32}{}^{M1}(Ca_{4.12}REE_{0.25})_{\Sigma4.37}{}^{M2}(\square_{1.16}Fe^{3+}{}_{0.90}Mn^{3+}{}_{0.89}Mg_{0.05})_{\Sigma3.00}{}^{M3}(\square_{0.51}Ti_{0.21}Nb_{0.19}Al_{0.09})$$
$$^{M4}(Si_{0.54}\square_{0.46})_{\Sigma1.00}{}^{Z1}(Zr_{3.00})Si_{24}O_{68.34}(OH)_{3.66}{}^{X1,2A}Cl_{1.17}{}^{X2A,B}(H_2O)_{1.42}$$

### 4.4. Crystal Structure: LV-117/226

*T-layers.* There are four independent Si1, Si2, Si3 and Si4 sites that are fully populated by Si. The additional M4 site has the refined occupancy of $Si_{0.42}$ and the <Si–O> distance of 1.568 Å. The octahedral M3 site was refined with the occupancy $Ti_{0.33}$ or 0.66 Ti apfu. The nine-coordinated N4 site has an average bond length of 2.694 Å and the refined occupancy of $Na_{0.91}Sr_{0.09}$.

*Z-layers*. The octahedral Z1 sites are fully populated by Zr. The average <Z1–O> distance is 2.071 Å. The N3 and N4 sites are situated within a distance of 0.80 Å and were refined with the occupancies of $Na_{0.75}$ and $Na_{0.25}$, respectively. The N3 site is coordinated by nine O atoms with the <N3–O> distance of 2.681. The N4 site is 8-coordinated with the <N3–O> distance of 2.575 Å.

*M-layer.* The octahedral M1A and M1B sites predominantly populated by Ca have site-scattering factors of 27.60 and 24.56 $e^-$ for the M1A and M1B sites, respectively. The REEs modeled using Ce Site-scattering curve were placed into the M1B site; its refined content of 0.36 Ceapfu is an excellent agreement with the chemical data (also 0.35 apfu). The refined occupancy of the M1A site is $Ca_{0.61}Zr_{0.39}$, and the total Zr-content (Z1 + M1A) is 4.15 apfu, in good agreement with the chemical data. The M1A and M1B sites have different mean bond lengths of 2.382 and 2.323 Å, respectively, and the polyhedral volumes of 16.17 and 15.61 $Å^3$, respectively. The M2 site is split into the M2A and M2B sites with the M2A-M2B distance of 1.465 Å. The M2A octahedral site has four short distances of 2.140 Å and two long distances of 2.588 Å. The refined occupancy of the M2A site is $Mn_{0.53}$. The refined occupancy of the low-occupied five-coordinated M2B site is $Fe_{0.11}$.

*X-sites*. The low-occupied X1A, X1B and X2A, X2B sites are populated by $H_2O$ molecules with a total refined content of 6.12 apfu. The X2C site was assigned to Cl with the total refined content of 1.73 Cl apfu.

The refined formula for the sample LV-117/226 can be written as:

$$^{N1-5}(Na_{9.76}Sr_{1.04})_{\Sigma 10.80}{}^{M1}(Ca_{4.49}Zr_{1.15}REE_{0.36})_{\Sigma 6.00}{}^{M2}(Mn^{3+}{}_{1.60}\square_{0.75}Fe^{3+}{}_{0.65})_{\Sigma 3.00}{}^{M3}(Ti_{0.66}\square_{0.33})$$
$$^{M4}(Si_{0.85}\square_{0.15})_{\Sigma 1.00}{}^{Z1}(Zr_{3.00})Si_{24}O_{72}{}^{X2C}Cl_{1.73}{}^{X2A,B}(H_2O)_{6.12}$$

The empirical formulae based on $Si_{24.85}$ may be written as:

$$^{N1-5}(Na_{8.31}Sr_{0.81}K_{0.16})_{\Sigma 9.28}{}^{M1}(Ca_{3.39}Zr_{1.25}REE_{0.33}Hf_{0.03})_{\Sigma 5.00}{}^{M2}(Mn^{3+}{}_{1.34}\square_{1.01}Fe^{3+}{}_{0.65})_{\Sigma 3.00}{}^{M3}(\square_{0.56}Ti_{0.21}Al_{0.15}Nb_{0.08})$$
$$^{M4}(Si_{0.85}\square_{0.15})_{\Sigma 1.00}{}^{Z1}(Zr_{3.00})Si_{24}O_{68.70}(OH)_{3.30}{}^{X2C}Cl_{1.34}{}^{X2A,B}(H_2O)_{4.66}$$

## 5. Discussion

An important feature of the geochemistry of the Lovozero massif, which distinguishes it from the neighboring Khibiny massif, is the low calcium content. The average calcium content in the massif rocks is 1.22 wt.%, and the Na/Ca ratio is 9.39 [32]. This is reflected in the chemical composition of rock-forming and accessory minerals. Ca-bearing minerals are not characteristic of the massif rocks [47], and the bulk of this element is included as an admixture in sodium-bearing minerals (e.g., rock-forming aegirine and magnesium-arfvedsonite).

EGM's crystallize in the rocks of the Lovozero massif during cooling foiditic magma at the temperature range 750–900 °C according to nepheline geothermometer data [48,49]. The geochemical behavior of Ca during crystallization of the Lovozero Massif rocks can be conventionally compared with the behavior of incompatible elements. In the early stages of rock crystallization, Ca was dispersed in rock-forming minerals (for example, as a diopside component in aegirine) and accumulated only by the later stages of crystallization. Ca sufficiently accumulates and forms its own minerals, for example, fluorapatite and titanite, only in the most evolved rocks. The EGMs require calcium for crystallization, 6 apfu of Ca are necessary for building an octahedral ring. Owing to the flexibility of the eudialyte structure, it can crystallize in conditions of extreme calcium deficiency. Thus, low-calcium eudialyte is formed, such as oneillite, raslakite and voronkovite. The calcium deficiency in these minerals is compensated by manganese and/or iron. Complex substitutions and the presence of extra Zr content with Ca-deficiency make some problems for the EGM normalization formula.

The normalization scheme based on Si + Al + Zr + Ti + Hf + Nb + Ta + W = 29 apfu proposed in [7] produces strong correlation between Zr and Si contents for the EGM samples from Mt. Alluaiv (Figure 5a). This correlation is misleading if the same graph is based on atomic amounts. The absence of correlation between the Si and Zr contents is

demonstrated in Figure 5b. The same problems with the normalization of the EGM formula were noted in [3]. These authors propose to use the atomic amounts of cations to trace changes in the composition of EGM during magmatic evolution. As an alternative way for the calculation of the EGM formula, we recommend a normalization scheme based on the Si content determined directly from the single-crystal XRD refinement for selected samples (two last columns in Table 1).

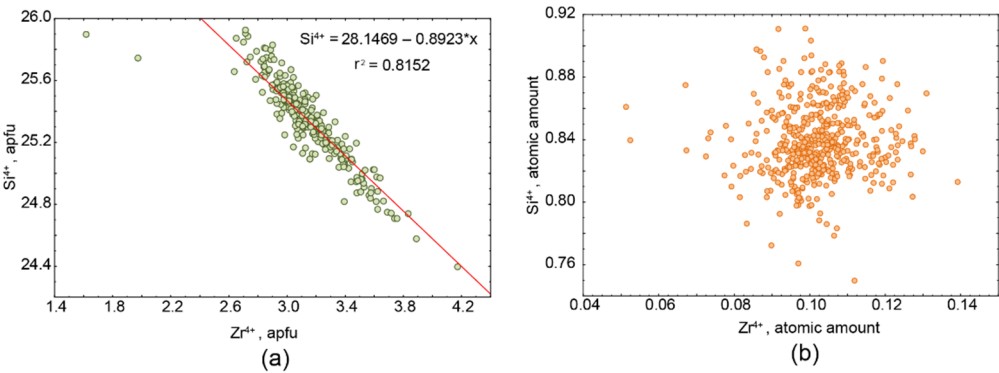

**Figure 5.** The relations between Si and Zr contents in EGM based on: (**a**) normalization scheme: Si + Al + Zr + Ti + Hf + Nb + Ta + W = 29 apfu, (**b**) atomic amounts.

The EGMs are usually distinctly zoned (Figure 2b) with cores that are Ca-deficient and Zr-enriched and marginal zones enriched by REE [3] (Table 1). Factor analysis of the data (Figure 6) on the composition of eudialyte from rocks of the eudialyte complex showed a decrease in Ca content with an increase in Zr content (as well as Al, Fe, Mg and Na). High Zr concentrations are observed in the cores of zonal eudialyte crystals from eudialyte lujavrites. The outer rims of these crystals are relatively enriched in calcium, as well as La, Ce, Ti, Nb and Mn. The same elements are usually enriched in homogeneous eudialyte grains from evolved rocks of the eudialyte complex, namely foyaites, porphyritic and fine-grained nepheline syenites. The Zr-rich EGMs are primary and possess no signs of secondary solid-state alteration [50].

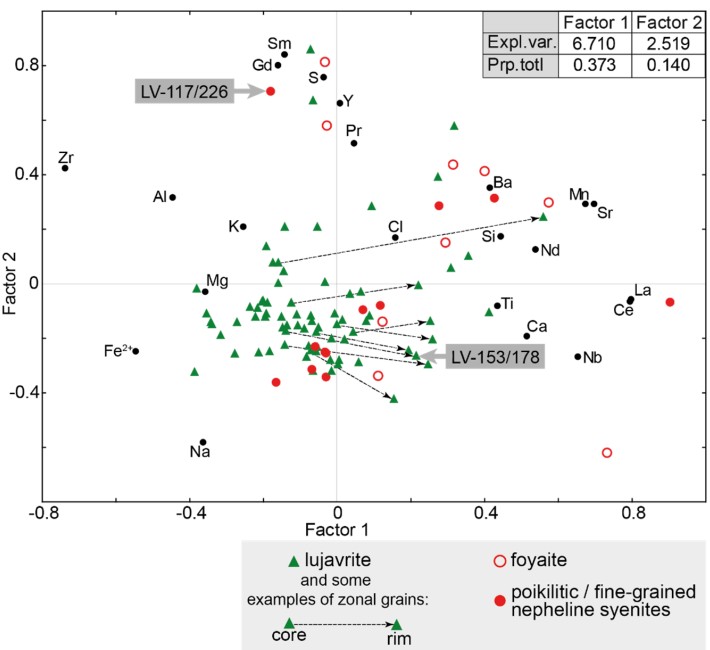

**Figure 6.** Results of factor analyses of data on the chemical composition of EGM from Eudialyte complex. Expl.var.—explained variance; Prp.totl—proportion of total variance. The meaning unit is factor score for both axis.

Forming of zonal EGM crystals with normal Ca content (6 apfu) at the late rims of EGM may be connected with (Figure 2b) Ca accumulation during crystallization of foiditic magma or as a result of partial melting/fenitization of Ca-rich xenoliths of Devonian volcaniclastic rocks [51].

The observed chemical variations in core and marginal zones of EGM well agrees with the observed two-phase and three-phase (davinciite–rastsvetaevite–"hydrorastsvetaevite") concentrically zoned crystals, where the central part is represented by davinciite, while the outer lighter rims are formed by rastsvetaevite and "hydrorastsvetaevite" [52,53].

According to our chemical data, the Zr content in EGMs may reach up to 4.2 apfu. The Zr amount linearly increases with the increasing Al content (Figure 7a). It seems that the incorporation of extra Zr may be connected with several complex substitution schemes, including vacancies and the possible incorporation of Al into the M4 site. The negative correlations with Zr were observed for REE, Ca and Nb (Figure 7b–d). It should be noted that the Zr-rich EGMs are depleted in Ca (Figure 7b), and this correlation is the most significant ($R^2 = 0.34$). In the crystal structures of EGMs, Ca and REE occupy the M1 octahedral site [1,7,8,15], and the incorporation of additional Zr is also connected with the M1 site. Nb occupies the octahedral M3 site [7]. The decrease in the Nb content correlated with the increasing Zr content may be connected with the vacancies at the M3 site.

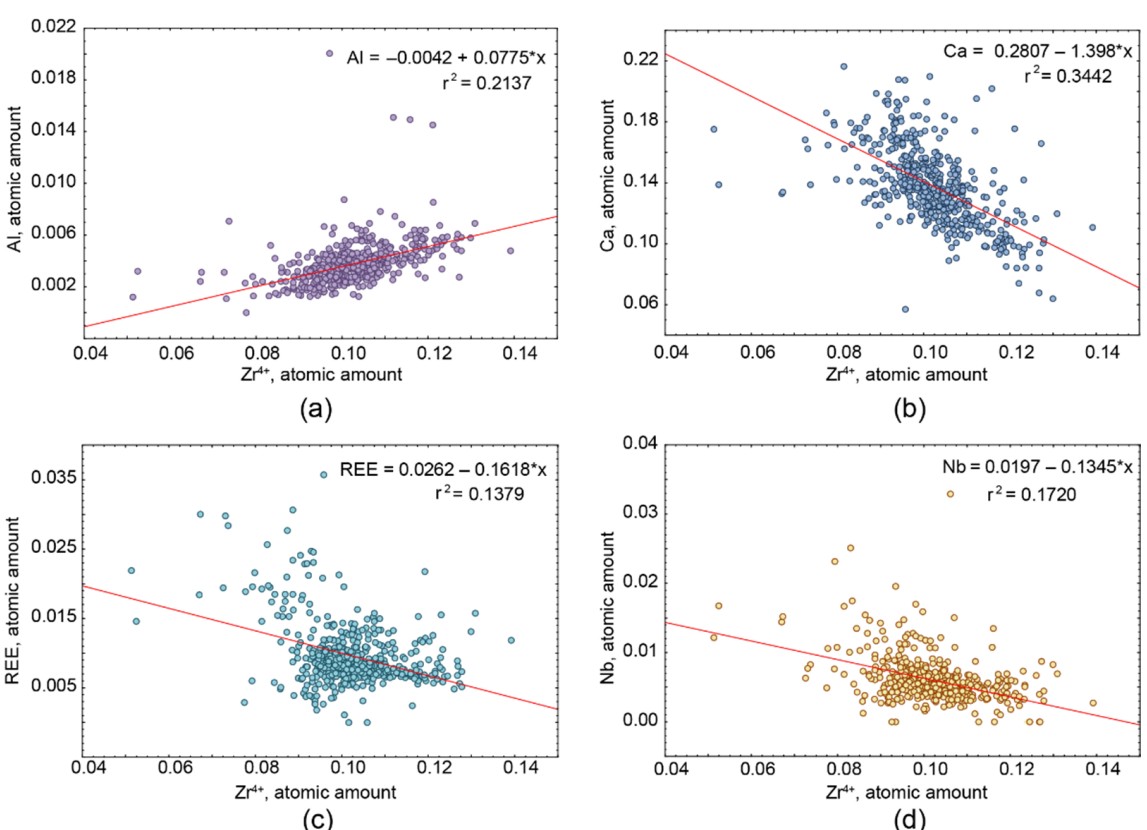

**Figure 7.** The relations between the cation contents in EGMs from Alluaiv Mt.: (**a**) plot $Al^{3+}$ vs. $Zr^{4+}$, (**b**) plot $Ca^{2+}$ vs. $Zr^{4+}$, (**c**) plot $REE^{3+}$ vs. $Zr^{4+}$, (**d**) plot $Nb^{5+}$ vs. $Zr^{4+}$.

The marginal zones of EGM grains with the small excess of Zr (0.2 apfu) or non-zonal grains with normal Zr-content (3 apfu) crystallize in the $R\bar{3}m$ space group with additional Zr placed into the five-coordinated M2A site according to the previous studies [46]. However, the M1 site contains excessive electron density (refined REE content is 0.54 apfu instead of 0.25 REE apfu from chemical data). It seems that extra Zr content is associated with the M1 site, which has more typical for Zr octahedral coordination. The five-coordinated Zr

(Figure 4) in inorganic structures is unlikely, which forces us to place all extra Zr into a more appropriate octahedrally coordinated M1 site.

According to the new data for Ca-deficient EGM from the Lovozero massif, the ordering of Mn into the M1 site leads to lowering of the total EGM symmetry to the *R*3 space group with the splitting of the M1 site into two subsites [54]. In the studied structure, the M1A and M1B sites differ in site-scattering factors (by more than 3 $e^-$) and in polyhedral volumes (16.17 vs. 15.61 Å$^3$). The incorporation of extra Zr into the M1A octahedral site is supported by the total refined Zr content (Z1 + M1A) of 4.15 apfu, which is in excellent agreement with the chemical data (Table 1). Such type of ordering results in local symmetry lowering for the six-membered octahedral ring and consequently lowering of symmetry from $R\bar{3}m$ in Zr-poor 153/178 sample to *R*32 in the crystal structure of Zr-rich LV-117/226 EGM sample (Figure 8).

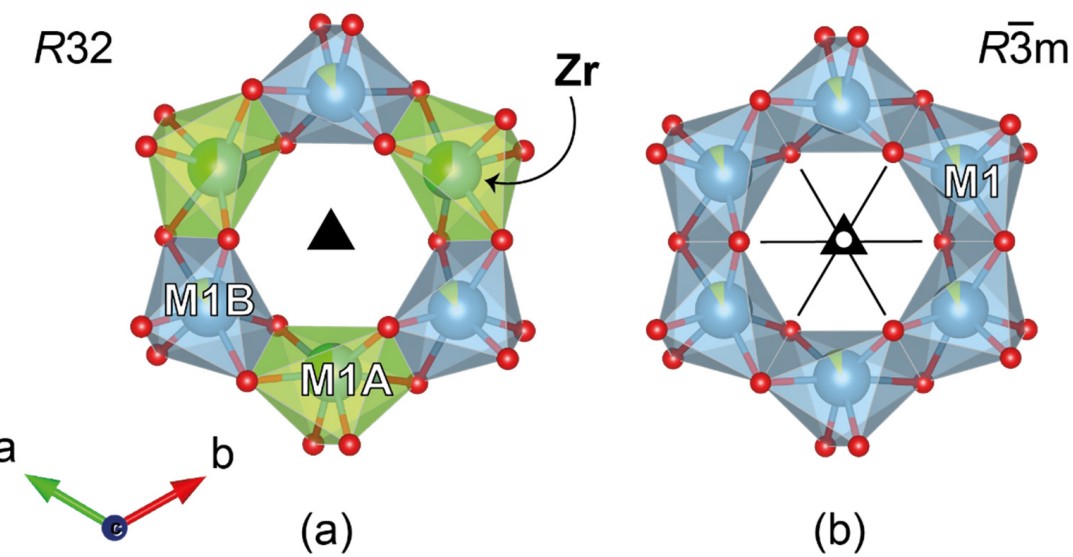

**Figure 8.** The splitting of the M1 site into the M1A and M1B sites in the crystal structure of LV-117/226 (**a**) and absence of splitting for 153/178 (**b**) eudialyte samples from Mt. Kedykvyrpakhk, Lovozero massif. Light-green octahedra represent mixed Ca/REE M1B sites, light blue—mixed Ca/Zr M1A (M1) sites. The local symmetry elements are indicated without a two-fold axis to simplify.

The complex substitution schemes in EGMs can be associated with the blocky isomorphism [55], which includes block-by-block substitutions involving groups of atoms with different coordination and topologies, and in the latest works are also called local heteropolyhedral substitutions [56]. This type of substitution usually involves the species-defining M2, M3 and M4 sites [1,8,57].

According to our structural data, at least one-half of the additional M4 sites are populated by H$_2$O. Summarizing our chemical (Figure 6) and structural data, we propose the possible ways of incorporation of extra Zr into the eudialyte structure via local heteropolyhedral substitutions:

$$^{M4}\square + {^{M1}}2Zr^{4+} \leftrightarrow {^{M4}}Si^{4+} + {^{M1}}2Ca^{2+}$$

The positive correlation between the Zr and Al contents in the EGMs from Mt. Kedykvyrpakhk (Figure 6) and their negative correlation with Ca and REE may be connected with the following complex substitution:

$$^{M4}Al^{3+} + {^{M1}}Zr^{4+} \leftrightarrow {^{M4}}Si^{4+} + {^{M1}}REE^{3+}$$

Both M3 and M4 sites are predominantly vacant and, taking into account the positive correlation between Zr and Al and the negative correlation with $Ca^{2+}$ and $Nb^{5+}$, the following complex substitution scheme can be proposed:

$$^{M3}\square + {}^{M4}Al^{3+} + {}^{M1}Zr^{4+} \leftrightarrow {}^{M3}Nb^{5+} + {}^{M4}\square + {}^{M1}Ca^{2+}$$

As already mentioned, 70% of the EGM samples from the rocks of the Lovozero massif are hyperzirconium, i.e., their Zr content exceeds 3 apfu [3]. Figure 9 shows a schematic section along the line I–II (see Figure 1b) and the distribution of elements in the composition of the eudialyte group minerals. The Ca (Figure 9a) and Zr (Figure 9b) are antagonists. The richest in zirconium are EGMs from eudialyte lujavrite. The highest Ca concentrations are characteristic of EGMs from foyaite and rocks of the poikilitic complex (leucocratic nepheline ± sodalite syenite) (Figure 9c). The Ca-enriched EGMs here have a normal Zr-content.

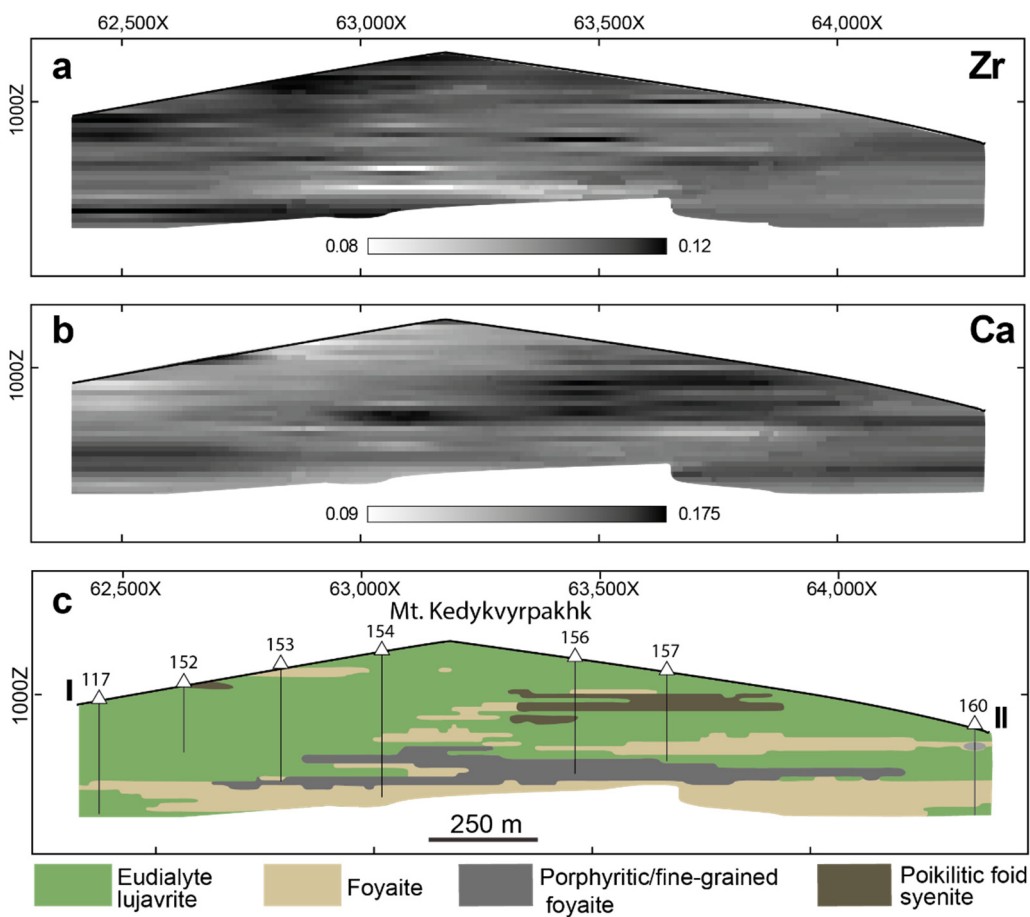

**Figure 9.** The section along the I–II line of Mt. Kedykvyrpakhk, Lovozero massif: (**a**) distribution of Zr (atomic amount) in EGM, (**b**) distribution of Ca (atomic amount) in EGM, (**c**) geology and drill holes. The cross-section was extracted from 3D block models generated by Micromine 2016.1. Interpolation was carried out by an inverse distance weighted method. The meaning units for X and Z axis are given in meters.

## 6. Conclusions

The EGMs, owing to their crystal structure flexibility, may be considered as geochemical indicators of crystallization conditions. The compositional variations in EGMs (strong zonation with cores enriched in Zr, Al and Ti and rims enriched in REE, Si, Ca, Sr and Mn) found in this study gives specific information about magmatic crystallization conditions in the Lovozero complex.

During the evolution of foiditic magma in Lovozero massif, Ca accumulated and crystallized at the last stages of EGM formation. The late EGM rims with normal Ca content connected with the process of Ca-accumulation during magma crystallization and/or addition Ca owing to melting/fenitization of Ca-rich xenoliths of Devonian volcaniclastic rocks.

The normal Zr content in EGM is 3 apfu, where Zr populates Z1 octahedral site. Most of the EGM samples (70%) from the rocks of the Eudialyte complex of the Lovozero massif are hyperzirconium, i.e., their Zr content exceeds 3 apfu. Additional Zr incorporates into eudialyte structure into the octahedral M1A site and replaces Ca with the symmetry lowering from $R\bar{3}m$ to $R32$.

There are three main substitution schemes associated with the incorporation of Zr into eudialyte crystal structure:

(1) $^{M4}\square + {}^{M1}2Zr^{4+} \leftrightarrow {}^{M4}Si^{4+} + {}^{M1}2Ca^{2+}$

(2) $^{M4}Al^{3+} + {}^{M1}Zr^{4+} \leftrightarrow {}^{M4}Si^{4+} + {}^{M1}REE^{3+}$

(3) $^{M3}\square + {}^{M4}Al^{3+} + {}^{M1}Zr^{4+} \leftrightarrow {}^{M3}Nb^{5+} + {}^{M4}\square + {}^{M1}Ca^{2+}$

**Supplementary Materials:** The following are available online at https://www.mdpi.com/article/10.3390/min11090982/s1, Table S1: Selected interatomic distances in the LV-153/178 EGM sample, Table S2: Anisotropic displacement parameters in the LV-153/178 EGM sample, Table S3: Selected interatomic distances in the LV- LV-117/226 EGM sample, Table S4: Anisotropic displacement parameters in the LV-117/226 EGM sample.

**Author Contributions:** T.L.P.: SC XRD experiments, statistics and original draft preparation. J.A.M.: material, writing—review and editing, visualization. A.O.K.: geostatistics, maps, review and editing. Y.A.P. and A.V.B.: BSE-images, electron microscope investigation. S.M.A. conceived of the work and reviewed the manuscript. S.V.K. conceptualization, review and editing manuscript. All authors discussed the manuscript. All authors have read and agreed to the published version of the manuscript.

**Funding:** This research was funded by Russian Science Foundation, project no. 20-77-10065.

**Acknowledgments:** Authors a grateful to the X-ray Diffraction Centre, Geo Environmental Centre "Geomodel" of Saint-Petersburg State University for experimental studies.

**Conflicts of Interest:** The authors declare no conflict of interest.

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
