# Peer review of "Zr-Rich Eudialyte from the Lovozero Peralkaline Massif, Kola Peninsula, Russia"

_minerals, doi:10.3390/min11090982_

Round 1

Reviewer 1 Report

The paper is a study on the crystal chemistry and structure of rare Zr-rich Eudyalite-group minerals from the Lovozero peralkaline layered complex, Russia, based mainly on EPMA and Single Cristal -XRD. In the magmatic complex, Zr-rich zoned EGMs are typical of eudyalite lujavrite rocks, while normal-Zr and homogeneous EGMs occur in foyaites and nepheline syenites. The analytical results document that Zr and Ca are antagonists in EGMs structures; the substitution of Zr for Ca and REE in the coordination sites of EGMs involves a structural adaptation that produces a lowering of symmetry of the crystals. Zoning and Zr/Ca substitutions in eudyalites of eudyalite lujavrite rocks probably mirror physicochemical variations in a general low-Ca environment. I found the research design and the methods appropriate, the discussion well argued and consistent with the experimental data; the conclusions are too concise, I suggest to expand them a little. English is generally good, I only found a short sentence to rephrase and some minor errors and typos to correct. See the attached file for my observations and suggestions.

Author Response

We are very grateful to reviewers for the constructive comments. We tried to revise our manuscript as best as we can.

Authors made grammar corrections as suggested reviewer in attached pdf-file. The additional text was added to the discussion and conclusion paragraphs:

Lines 323-328:

EGM’s crystallize in the rocks of the Lovozero massif during coolling foiditic magma at the temperature range 750-900 C° according to nepheline geothermometer data [48,49]. Geochemical behavior of Ca during crystallization of the Lovozero Massif rocks can be conventionally compared with behavior of incompatible elements. In the early stages of rock crystallization, Ca was dispersed in rock-forming minerals (for example, as a diopside component in aegirine), and accumulated only by the later stages of crystallization.

Lines 446-455:

The EGMs owing its crystal structure flexibility may be considered as geochemical indicators of crystallization conditions. The compositional variations of EGMs (strong zonation with cores enriched in Zr, Al, and Ti and rims enriched in REE, Si, Ca, Sr, and Mn) found in this study gives specific information about magmatic crystallization conditions in the Lovozero complex.

During evolution of foiditic magma in Lovozero massif, Ca accumulated and crystallized at the last stages of EGM formation. The late EGM rims with normal Ca content connected with process of Ca-accumulation during magma crystallization and/or addition Ca owing to melting/fenitization of Ca-rich xenoliths of Devonian volcaniclastic rocks.

Reviewer 2 Report

The manuscript of Panikorovskii et al. is solid and very informative work concerning Zr-rich eudialytes from the Lovozero peralkaline massif, Kola Peninsula. The work is well organized, and it will be of interest to the mineralogical community. I have just a few minor remarks.

Figure 2 caption. It should be “Backscattered”. The authors should include comments concerning the reason for the light and dark contrasts in the BSE and relate them with the properties of the assigned minerals. For example, why is EGM lighter than Ab, Mc, and Nph?

Figure 2b. More details should be added to the petrographic analysis. Please provide identification criteria for each mineral. It would be nice if the chemical zonation in the crystals could be related to changes in geologic conditions and the sequence of mineral crystallization.

Figure 9. The authors should indicate the meaning (units) of the X and Y axes.

Author Response

 We are very grateful to reviewers for the constructive comments. We tried to revise our manuscript as best as we can.

Comment

Figure 2 caption. It should be “Backscattered”. The authors should include comments concerning the reason for the light and dark contrasts in the BSE and relate them with the properties of the assigned minerals. For example, why is EGM lighter than Ab, Mc, and Nph?

Response:

Figure caption were corrected as suggested reviewer. Brightness of mineral in BSE images depends on average mass (mean atomic number) of its constituent elements: the heavier mass corresponds to the brighter color. For the albite, NaAlSi3O8 mean atomic number is 10 ((11+13+14*3+8*8)/13 = 10, for nepheline, Na3K(Al4Si4O16) is 10.28, for microcline, KAlSi3O8 is 10.62 and for EGM, Na15Ca6Fe3Zr3Si(Si25O73)(O3)Cl2 is 11.37. So 11.37 > 10.62 > 10.28 > 10 and EGM brighter > microcline > nepheline > albite.

Comment

Figure 2b. More details should be added to the petrographic analysis. Please provide identification criteria for each mineral. It would be nice if the chemical zonation in the crystals could be related to changes in geologic conditions and the sequence of mineral crystallization.

Response

The next optical characteristics were used for mineral determination via Leica M205 polarizing stereomicroscope: Optical relief, birefringence, cleavage, fracture, transparency, colour, pleochroism, surface relief. All minerals also diagnosed by the scanning electron microscopy (see added section Materials and Methods, Lines 139-142:)

The thin polished sections were analyzed using the scanning electron microscope LEO-1450 (Carl Zeiss Microscopy, Oberkochen, Germany), with energy-dispersive microanalyzer Quantax 200 to obtain BSE (Back Scattered Electron) images and pre-analyze all detected minerals.

Comment

Figure 9. The authors should indicate the meaning (units) of the X and Y axes.

Response

The X and Z given in meters, information added to the figure caption (Line 449).

Reviewer 3 Report

The article is interesting. It is dedicated to minerals of the eudialyte group, distinguished by their complexity and diversity of composition. Zonal distribution is shown: zirconium in the center of crystals, rare earth elements along the periphery. This is very important from a practical point of view, since minerals of the eudialyte group can be an important source of these critical elements. The high quality and informativeness of the illustrations should be noted.

Flaws:

  1. A number of eudialyte deposits are mentioned in the review, but the level of zirconium and rare earth elements in eudialyte of different deposits remained unclear.
  2. It remains unclear which minerals from the eudialyte group are the most enriched with zirconium and rare earth metals.
  3. Despite the fact that eudialyte is a group of minerals, and the authors studied in detail the composition of minerals, they only mention the name of specific minerals in the samples studied in one place.
  4. Despite the fact that a number of the cited works were published in Russian, there is no corresponding indication for these works in the list of used literature. 

Author Response

 We are very grateful to reviewers for the constructive comments. We tried to revise our manuscript as best as we can.

Flaws:

1. A number of eudialyte deposits are mentioned in the review, but the level of zirconium and rare earth elements in eudialyte of different deposits remained unclear.

The information about ZrO2 and REE2O3 content of EGM from different deposits were added to the text.

Response:

Additional information was added to the text. Lines 62-70.

According to literature data the Zr-content in EGM from most deposits does not exceed 3 apfu. The ZrO2 and REE2O3 content varies in the range (wt.%): 11.40 − 12.07; 2.00 − 3.30 for North Qôroq centre, South Greenland, 10.35 − 11.48; 0.39 − 10.15 for Mont Saint-Hilaire, Quebec, Canada, 9.85 − 10.82; 0.15 − 0.52 for Gardiner complex, East Greenland, 10.90 − 11.20, 1.15 − 12.12 for Ilímaussaq, Southern Greenland, 0.15 − 5.00 (ZrO2) for Tanbreez, South Greenland [7,26–28]. Meanwhile, most of the EGM samples (70%) from the rocks of the Lovozero massif are hyperzirconium, i.e., their Zr content exceeds 3 apfu [3]. According to our data the ZrO2 content in EGM from Lovozero massif ranging from 6.32 − 17.14 wt.% and REE2O3 varies in the range 0.35 − 5.92 wt.%.

2. It remains unclear which minerals from the eudialyte group are the most enriched with zirconium and rare earth metals.

Response:

The Zr-rich EGM (17.14 wt.% of ZrO2) according our data should be manganeudialyte. But Si content (24.4 apfu) lower than in manganeudialyte and possibly may coorespond to the separate mineral species (not approved by IMA CNMNC)

3. Despite the fact that eudialyte is a group of minerals, and the authors studied in detail the composition of minerals, they only mention the name of specific minerals in the samples studied in one place.

Response:

The EGM usually represented by (from often to rare): manganeudialyte, eudialyte, kentbrooksite and alluaivite. It should be noted that exact determination of mineral species may performed only with SC XRD, EMPA, Mossbauer and TG-DSC studies.

4. Despite the fact that a number of the cited works were published in Russian, there is no corresponding indication for these works in the list of used literature. 

Response:

All works originally in Russian was marked in the reference list as (in Russian)

Reviewer 4 Report

The manuscript presents two cell refinements of EGM in Lovozero. The methods are classical and crystallographic data are correctly presented. Some typos were detected and editing is mandatory.

Minor revision is requested.

Author Response

 We are very grateful to reviewers for the constructive comments. We tried to revise our manuscript as best as we can.

All detected typos was fixed in the revised version of the manuscript.